

# English teaching system design based on ID3 algorithm and node optimization

Wangmeng Jiang[1] and Qiang Ma[2]

[1] Faculty of Humanities and Social Sciences, Beijing University of Technology, Beijing, China
[2] Faculty of Department of Foreign Language and Tourism, Hebei Petroleum University of Technology, Chengde, Hebei, China

## ABSTRACT

In order to optimize the integration of English multimedia resources and achieve the goal of sharing English teaching resources in education, this article reconstructs the traditional college English curriculum system. It divides professional English into learning modules according to different majors integrating public health teaching resources. How optimize the integration of English multimedia resources and achieving the goal of sharing English teaching resources (ETR) is the main direction of English teaching reform during the current COVID-19 pandemic. An English multimedia teaching resource-sharing platform is designed to extract feature items from multimedia teaching resources using the ID3 information gain method and construct a decision tree for resource push. In resource sharing, a structured peer-to-peer network is used to manage nodes, query location and share multimedia teaching resources. The optimal gateway node is selected by calculating the distance between each gateway node and the fixed node. Finally, a collaborative filtering (CF) algorithm recommends Multimedia ETR to different users. The simulation results show that the platform can improve the sharing speed and utilization rate of teaching resources, with maximum throughput reaching 12 Mb/s and achieve accurate recommendations of ETR.

# INTRODUCTION

In the contemporary epoch of public health, the domains of information technology and Internet platforms have delved profoundly into the realm of education and the pedagogical milieu. Within the sphere of English education, the focal point lies in the art of effectively guiding diverse learners towards the utilization of these resources, thereby enabling them to harmoniously amalgamate their individual requirements and augment the applicability of supplementary pedagogical materials. Concomitantly, the predicament of "resource overload" incessantly vexes educational institutions, thereby engendering an exigency for the expeditious implementation of a cutting-edge intelligent recommendation algorithm. This algorithm shall serve the purpose of tailor-made recommendations for distinctive English teaching resources (ETRs) predicated upon the proclivities and idiosyncrasies of

Corresponding author
Qiang Ma, cdpc_mq@cdpc.edu.cn

English learners, as well as pertinent circumstantial factors, thereby culminating in an amplified efficacy (*Gao, Xing & Yin, 2021*; *Yang, 2021*).

Vocational education should improve the quality of workers, especially their professional ability. In higher vocational colleges, students' majors are closely related to vocational production activities. English is no longer primary language teaching but is combined with students' majors to solve professional problems in English, highlighting its applicability (*Xie, 2021*). English language education is bifurcated into two distinct categories: general and specialized English language education. Consequently, the English language resource library must comprise both rudimentary material and content about different professional domains: available and professional English. It would meet the needs of tertiary educators, business professionals, corporate trainers, and language aficionados, enabling them to improve their information retention and professional capacities. It would support students' specific interests, admission exam requirements, and employment requirements while creating ideal conditions for self-directed learning (*Tan & Shao, 2021*).

Presently, numerous institutes of higher education have orchestrated collaborative efforts between English instructors and educational technicians to effectuate the amalgamation of teaching resources, fashion multimedia teaching courseware, cultivate online courses, and cultivate an optimal internal milieu conducive to the seamless exchange of pedagogical materials (*Juan & Yahaya, 2020*; *Wang & Chen, 2020*; *Ansari & Khan, 2020*). Nevertheless, the prevailing multimedia resource-sharing platforms have proven to be beset by disorderliness, consequently burdening educators and learners with redundant tasks and neglecting to furnish them with high-caliber channels for acquiring resources, thereby impeding the augmentation of students' self-directed learning proficiencies. Consequently, scholars and researchers have advanced proposals for ameliorating the English multimedia resource-sharing platform. *Wang & Qiao (2020)* has extensively employed blockchain technology in the domain of ETR sharing. Through a meticulous analysis of algorithmic efficacy, this article proffers an English teaching resource-sharing platform comprising key constituents such as the presentation layer, business layer, and data layer. Regrettably, the construction of said platform entails a considerable degree of complexity, thereby engendering superfluous squandering of human and material resources. In a similar vein, *Tarus, Niu & Kalui (2018)* has posited a resource-sharing platform underpinned by an enhanced collaborative recommendation algorithm, which harnesses hybrid recommendation algorithms to facilitate the dispensation of learning resources. Alas, this approach overlooks the imperative aspect of platform security performance, rendering it susceptible to pernicious interference.

The collaborative filtering algorithm stands as a quintessential and widely employed recommendation algorithm, perpetuating its legacy since its inception in 1992. Its fundamental premise revolves around uncovering users' interests by means of data mining their historical behavioral patterns, subsequently classifying users based on their distinct preferences, and ultimately recommending products bearing resemblance to their specific interests. Collaborative filtering algorithms can be broadly categorized into two types: User-CF and Item-CF, with most algorithms falling exclusively into one of these categories. However, this study takes into account the disparities between teaching resources and

users' registration time. The development of a shared ETR platform for vocational colleges caters to the day-to-day requisites of Chinese higher education institutions, fostering an elevation in the standard of English instruction, while concurrently furnishing students with a plenitude of high-quality, diverse resources, and bespoke services to bolster their autonomous learning endeavors. Constructing a public platform has the potential to enhance the quality of vocational education personnel training and bolster social service capabilities, thereby bearing immense practical significance. The primary contributions of this study encompass:

(1) Accomplishing the proficient classification of English resources through the utilization of the ID3 algorithm, which involves the consideration of eight attributes, namely, question type, subject, difficulty, reading ability, cognitive style, learning objective, learning situation, and learning effect, with their conditional entropy duly calculated;

(2) Revamping the conventional college English curriculum system by devising an English multimedia teaching resources sharing platform, and employing collaborative filtering algorithms to provide personalized recommendations of multimedia English teaching resources to diverse users.

(3) Effectively facilitating the sharing of multi-node networked intelligent teaching multimedia resources, ensuring rapid and precise multimedia dissemination.

## COLLEGE ENGLISH TEACHING RESOURCE LIBRARY

College English courses are divided into basic English and professional English by reconstructing the traditional curriculum system. Professional English is then further divided into various learning modules according to various majors, such as tourism English (*Ho, 2020*), business English (*Tratnik, Urh & Jereb, 2019*), accounting English (*Nuraini, Hanafiah & Lubis, 2020*), *etc*. There are many different varieties of professional English, and each module's difficulty level is designated according to learning requirements and the general public's needs. Each module includes curriculum standards, instruction tips, instructional resources, and questions for associated exercises. Learners can choose the appropriate learning content based on their current proficiency level and need by creating a curriculum centre, a training centre, a certificate examination centre, an employment service centre, and a communication platform. It can help build a solid language foundation.

Additionally, it can develop their capacity for real-world application. Figure 1 depicts the overall layout of the college ETR database. College English has strong pertinence, including basic English knowledge, relevant professional terms, and communication cases. Therefore, from the resource-sharing platform of higher vocational education, the construction of a higher vocational English resource database generally includes a material database, course resource database, question database, case database, video database, expanded resource database and so on *Li (2020)*. The following focuses on the course resource database, case database, and expanded resource database in detail.

The database of course resources primarily contains the lesson plan, vocabulary information, excellent course materials, micro-class course materials, teaching techniques,

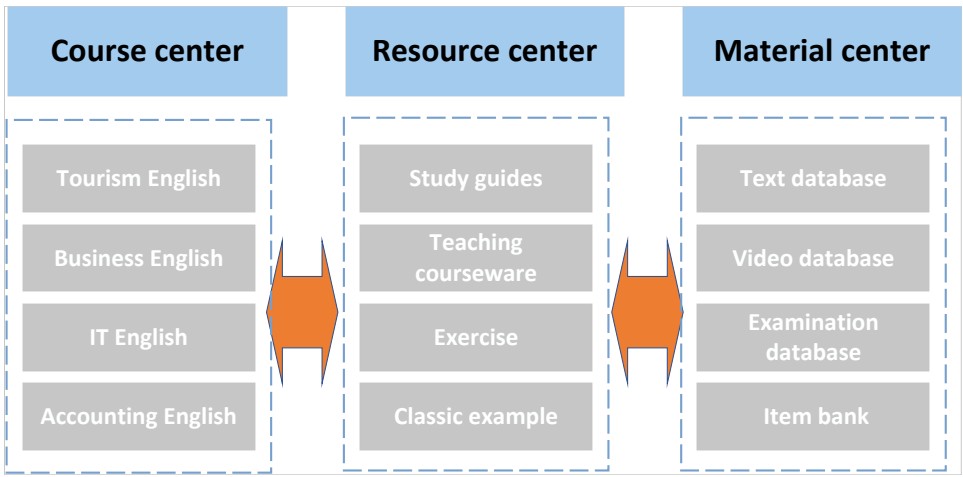

**Figure 1** College English teaching resource library.

*etc.* The foundational courses for English majors are integrated. In addition to written content, it contains videos, images, and other things. Excellent training materials will explain important and challenging concepts, answers to common problems, business jargon, *etc.* The school's intranet must be updated with the library's educational materials so teachers and students can share them (*Du, 2021*).

Most cases in the case database are good practice cases for students and career cases taken from reputable newspapers, periodicals, websites, and foreign language instruction. Students can simulate and study professional cases through virtual simulation technology and occupation-related real scenarios in addition to learning excellent cases better to comprehend professional knowledge and work tasks in social reality.

The expanded resource database includes English news, various English competitions, and professional-related development materials. The resource database can provide students with extra-curricular knowledge, broaden their horizons, understand the background culture, and improve their humanistic quality. It can also realize the cultivation of cross-cultural skills and strengthen their professional ability.

## ENGLISH MULTIMEDIA TEACHING RESOURCE SHARING PLATFORM

### Multimedia teaching mode

In the school teaching process, the integration and new language learning process have been entered through multimedia facilities. As shown in Fig. 2, the primary implementation mode of the system.

Through the above system functions, students' learning methods can be changed. Using multimedia and the Internet has changed the classroom environment, making it more engaging for pupils. In addition to providing students with a plethora of material, such as vocabulary, visuals, pronunciation, and cultural language, video education exposes students to real language in the classroom (*Peng, 2019*).

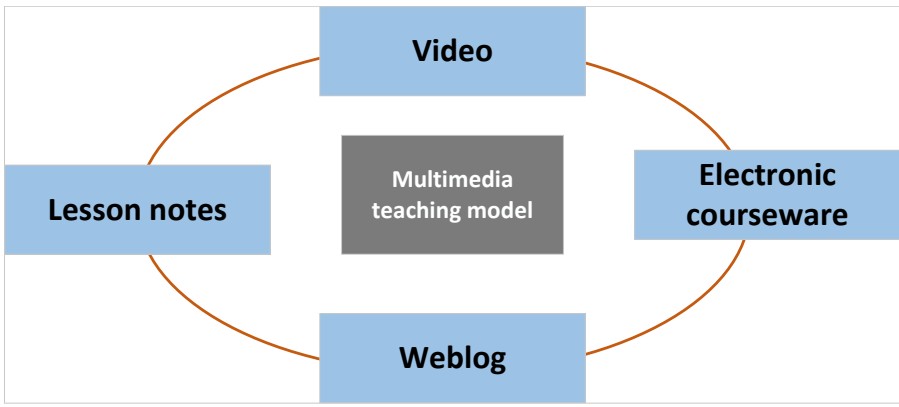

**Figure 2**  **Multimedia English teaching mode.**

## Resource sharing
### *Feature extraction*
The digital teaching resources stored in the computer hard disk are collected and processed through the system's front-end collecting equipment. The resource processing process is as follows: first, delete the processing. Delete duplicate and shortage of valuable resources; avoid occupying storage space; improve the system's running speed; delete batch based on JSP pages; and second, word processing. English text in the tense verb changes and irregular changes in the plural number of nouns promote the word to appear in a variety of forms, so in the word processing, the same word in different forms will be treated as different words, resulting in complex feature items.

English multimedia teaching resources are characterized by an extensive vocabulary, necessitating feature extraction, *i.e.,* the automated selection of keywords capable of representing content from these resources. This module aims to filter out words with low or no information, simplify the vector space dimension calculation process, avoid overfitting, and simplify calculations while improving classification accuracy (*Yang, 2022*). There are several techniques for extracting lexical features, among which this article employs the ID3 information gain method to extract lexical features.

Initially, the characteristics of the learners, resource features, and classification attributes are extracted and stored in a two-dimensional array. Subsequently, the discrete data is subjected to preprocessing to achieve normalization. During the process of adaptive resource push, eight attributes warrant consideration: question type (Th), subject matter (Qt), difficulty (Di), reading ability (Ra), cognitive style (Cs), learning goal (Lt), learning situation (Ls), and learning effect (Re). Each attribute possesses a distinct range of values. The conditional entropy of each attribute is computed individually, along with the information entropy of the remaining attributes under the condition of the classification category (*Devasenapathy & Duraisamy, 2017*).

Taking attribute Th as an example, based on Eq. (1), the information entropy generated by attribute Th is given by:

$$
\begin{aligned}
\text{Entropy}_{\text{Th}}(X) &= \sum_{i=1}^{k} \frac{|X|}{|X|} \text{Entropy}(X_i) \\
&= \frac{|X_{Sc}|}{|X|} \text{Entropy}(X_{Sc}) + \frac{|X_{Ns}|}{|X|} \text{Entropy}(X_{Ns}) \\
&= \frac{n_c}{n} \left[ -\frac{n_m}{n_c} \log 2 \left( \frac{n_m}{n_c} \right) - \frac{n_n}{n_c} \log 2 \left( \frac{n_n}{n_c} \right) \right] \\
&\quad + \frac{n_s}{n} \left[ -\frac{n_i}{n_s} \log 2 \left( \frac{n_i}{n_s} \right) - \frac{n_j}{n_s} \log 2 \left( \frac{n_j}{n_s} \right) \right].
\end{aligned}
\tag{1}
$$

The information gain of Th attribute is shown in Eq. (2):

$$
\text{Gain}(\text{Th}) = \text{Entropy}(X) - \text{Entropy}_{\text{Th}}(X).
\tag{2}
$$

In this way, the value of each node is used to construct the maximum value of the multimedia resource.

### Resource sharing process

Through the resource sharing module, the system provides users with remote use of resources on the machine, reducing the waste of resources in the process of sharing. Figure 3 shows the sharing process of the whole English multimedia teaching resource-sharing platform.

In the process of peer-to-peer resource sharing, the peer-to-peer resource sharing platform is used to achieve the goal of the process. There are three nodes in the module. Users can select nodes to perform tasks at any time (*Rahman, Newaz & Au, 2020*).

First, each node can play two roles at the same time, which are resource users and resource providers. Other remote nodes can run jobs of this node, and this node can also execute jobs of other remote nodes.

Then, when a node is a submission node, it can detect, access special editors, and count reputation. These reports exist in peer-to-peer networks and can be accessed and viewed by all nodes (*Sun, Yu & Fan, 2020*).

Finally, when a node assumes the role of an execution node, it is equipped with a reliable Java Virtual Machine (JVM) that facilitates the transmission of application progress information to reports.

### Node optimization

The distance between each gateway node and the fixed node is calculated. According to the distance, whether the gateway node has stability can be judged. The formula of the weighted average method is as follows:

$$
G_a = \alpha H_{na} + \beta \text{LQ}_{na} + \theta E_{na} + \varepsilon \text{LD}_{na} + \delta \text{SY}_{na}.
\tag{3}
$$

Where Eq. (3) shows that: $G_a$ represents the node selection factor, which is used to describe the selected node $a$ that can be regarded as a comprehensive measure of a gateway;

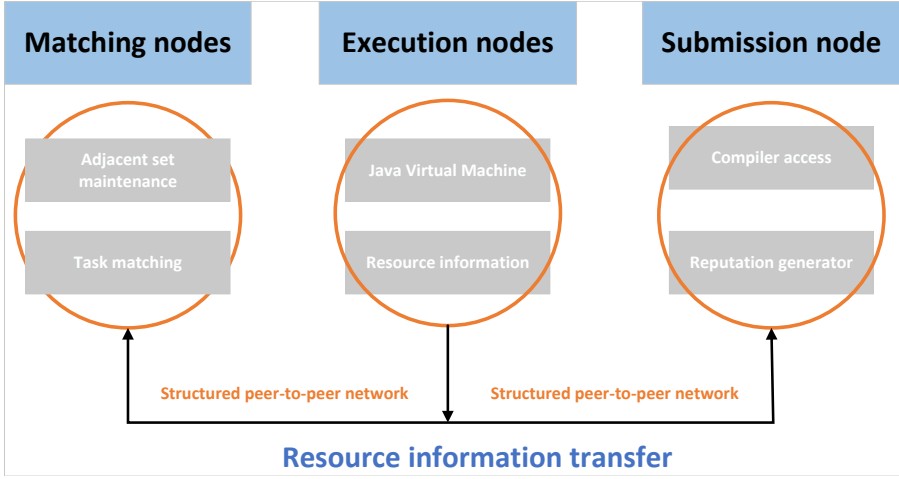

**Figure 3  Resource sharing process.** In the process of peer-to-peer resource sharing, the peer-to-peer re-source sharing platform is used to achieve the goal of the process.

node $n$ to gateway node $\alpha$ is defined as $H_{na}$, the weight is expressed as $\alpha$; node $n$ to gateway node $\alpha$, the link quality is defined as $\text{LQ}_{na}$, and the weight is expressed as $\beta$; It should be noted that the overall performance of a node is inversely proportional to its selection factor, implying that a smaller selection factor indicates a higher level of comprehensive performance.

The stability of the primary gateway nodes is contingent upon their associated main gateway nodes and the distance from node n. A greater degree of stability in the fluctuation of distance signifies a superior level of stability in the gateway node. Utilizing the secondary gateway node as the gateway facilitates the expeditious and precise distribution of multimedia teaching resources.

$$\text{SY}_{na} = D_t^{n,a} = \frac{1}{n}\sum_{m=0}^{n-1}\left\{d_{t-m}^{n,a} - E\left(d_t^{n,a}\right)\right\}^2. \tag{4}$$

The average distance between node $n$ and gateway node $a$ at time $t$ is shown in Eq. (5):

$$E\left(d_t^{n,a}\right) = \frac{1}{n}\sum_{m=0}^{n-1}d_{t-m}^{n,a}. \tag{5}$$

The selection process entails identifying the value with the most minimal variance, indicating the optimal gateway node. This node is then designated as the gateway exit, enabling access to the external network through this node. By employing the node selected at this juncture, the intelligent sharing of multimedia teaching resources and the attainment of enhanced speed and precision in multimedia sharing across multiple nodes can be accomplished within the network.

## Resource recommendation

Within the multimedia ETR resource-sharing platform, various types of resources are encompassed, including images, audio files, videos, and more. Addressing the limitation

of neglecting semantics within the collaborative filtering recommendation algorithm, *Wang & Fu (2021)* introduced a knowledge graph framework to embed semantic data into a lower-dimensional semantic space. This approach effectively compensates for the deficiencies of the collaborative filtering algorithm at the semantic level by computing semantic similarity. To tackle the challenge of feature value fusion, *Yang & Tan (2022)* leveraged deep learning techniques to integrate diverse features, proposing a personalized recommendation algorithm based on knowledge graph learning and ranking. *Wang et al. (2021)* focused on resolving the issue of students' knowledge loss within the learning system. They analyzed the characteristics of learners' needs, employed a knowledge graph to manage learning resources, and presented an architecture for personalized learning resource recommendations based on the knowledge graph. In an effort to enhance the accuracy and user satisfaction of the recommendation system, *Wan & Niu (2019)* employed a knowledge graph in three key areas: ontology recommendation generation, open linked data utilization, and graph-embedded recommendation generation. This comprehensive approach addresses the challenges associated with recommendation system accuracy and user satisfaction.

Hence, utilizing the traditional association rules algorithm to recommend these contents may not readily facilitate cross-domain recommendations (*Zhang, Ni & Zhao, 2014*; *Song & Li, 2022*). Furthermore, ETRs typically comprise attributes such as the primary name, resource type, content introductions, author profile, and upload time. On the other hand, student attributes primarily include student numbers, primary code, education type, and other relevant characteristics. During the recommendation process, it is also taken into account that students may express an interest in teaching resources pertaining to other majors apart from their own.

After using teaching resources, different ETRs will be evaluated. Suppose that the set of user groups is represented as $U = \{U_1, U_2, \ldots, U_m\}$, resource item set $D = \{D_1, D_2, \ldots, D_m\}$;

Among them, $m$ is the number of behavioural programs for teaching programs, $m$ represents the number of users, $U(m, n)$ represents the user $U_m$ is relative teaching resources $D_i$ or the degree of interest.

To build users $U_i$ and $U_m$ should find the evaluation resources of the two to score respectively to form the teaching resources collection $I_{im} = I_i \cup I_j$; For some ungraded items, the similarity between ungraded items and graded items is calculated to predict the scoring results of the ungraded items. Repeat the process until the unscored items are filled.

In the conventional process of computing similarity, the unrated items in the user-item rating matrix are typically substituted with a value of 0. However, this approach poses a challenge. For instance, if users only evaluate a fraction, say less than 1%, of the items, then the vast majority—over 99%—of items remain unrated. Consequently, computing similarity between these items yields similarity scores surpassing 98%. Clearly, such outcomes fail to elucidate the true similarity between users. To address this issue, when dealing with projects involving ungraded teaching resource items, an alternative approach is adopted. It leverages the similarity of other users to make predictions, employing the Pearson similarity calculation method (*Zhang, Zhou & Sun, 2017*).

## EXPERIMENT AND ANALYSIS

### Test environment

To prove the performance reliability of the platform in this article, taking the P2P-BT platform as reference (*Ab Wahid et al., 2015*), it is deployed on virtual hardware with the same hardware level as the built platform.

### Platform test

In this article, the virtual host throughput is not affected by the virtual load of about 5 min, as shown in Figs. 4 and 5.

The figure illustrates that the platform attains the highest throughput when multimedia resources are shared, with a maximum achievable throughput of 12 MB/s. This indicates that the sharing speed of English multimedia teaching resources within this platform is notably faster, rendering it highly suitable for online English teaching. However, it is worth noting that the phenomenon of concurrent transmission causing network congestion leads to spikes in request–response time for the P2P BT platform. As a result, the platform encounters challenges in processing a large number of client requests, leading to extended response times for resource sharing. Despite these limitations, utilizing the P2P BT platform in the teaching environment yields satisfactory outcomes, making it an ideal choice for sharing English multimedia teaching resources.

### Recommended results of ETR

Using the dataset derived from user interactions on the English multimedia teaching resource-sharing platform, a random sample of 2000 user records was selected for analysis. This selection aims to evaluate the effectiveness of the hybrid recommendation algorithm in achieving the desired recommendation outcomes. In this experiment, we examined the recommended effect of various recommendation models and compared the precision and recall rates of four models (*Debat, Speir & En Lin*) of User-CF, Item-CF, CB, and HybridR. Figure 6 displays the comparison outcomes of various recommendation methods.

User collaborative filtering has the lowest accuracy, recall, and F1 values, and its recommendation effect is subpar. The hybrid recommendation algorithm optimizes the issues of cold start and long tail and enhances the model's accuracy, so the recommendation effect is better than the other three algorithms. This is due to the weighted mixing of Item-CF and CB. Different teaching resource types can be recommended to appropriate users to satisfy the functional requirements of various role users for the system, thereby increasing the sharing and utilization rate of ETR. This is accomplished by integrating the collaborative recommendation algorithm into the teaching resource management system.

College English courses work to further expand and enhance the creation of three-dimensional teaching resources as teaching concepts are deepened. Promote the deep integration and practical application of teaching resources and English classrooms, enhance students' autonomous learning ability and comprehensive literacy, actively develop and improve teachers' comprehensive teaching abilities, fully utilize the "instrumental" and "humanistic" roles of college English, and contribute to promoting the comprehensive development of application-oriented universities.

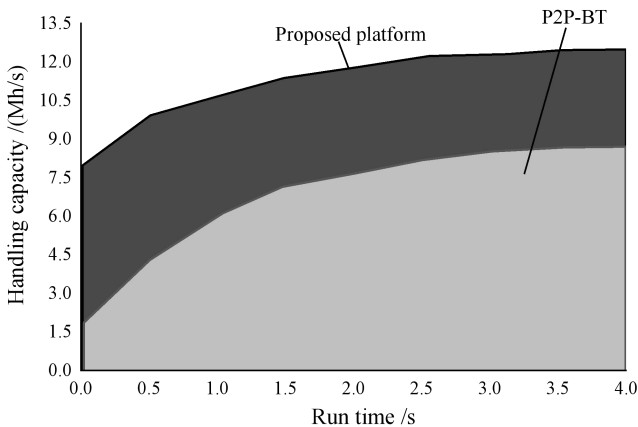

**Figure 4 Throughput of English multimedia teaching resource sharing platform.** The throughput of this platform is the highest when sharing multimedia resources, and the maximum throughput that can be achieved is 12 MB/s.

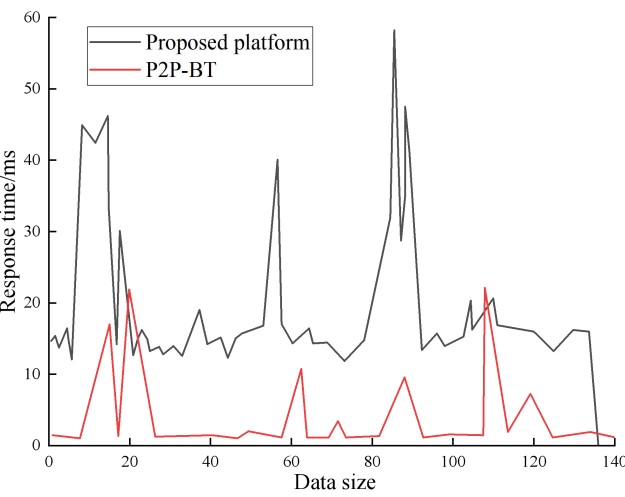

**Figure 5 Response time of English multimedia teaching resource sharing platform.** The request response of P2P-BT platform will increase sharply at a certain stage, which is due to the phenomenon that the network is blocked by concurrent transmission.

## CONCLUSION

Current research and thinking in the field of computers and education center around enhancing the extraction of educational information to enhance the efficacy of resource data sharing. The findings reveal that the proposed platform outperforms the comparative platform in terms of efficiency and network throughput. Through optimization of similarity calculations, the hybrid recommendation algorithm demonstrates improved model accuracy, recall rate, and F1 value. By utilizing this platform, users can effectively harness campus-wide resources within the classroom, maximizing the utilization of teaching

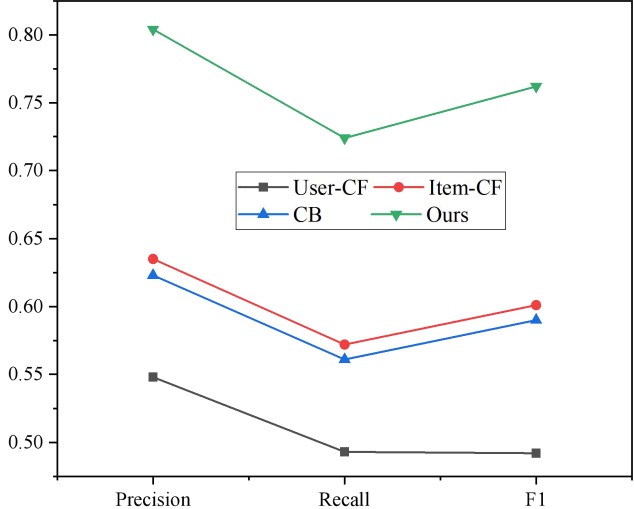

**Figure 6** **Comparison results of different recommendation algorithms.** The accuracy rate, recall rate and F1 value of user collaborative filtering are the lowest, and the recommendation effect is poor.

multimedia resources while minimizing waste. Moreover, the platform offers real-time on-demand teaching and facilitates evaluation of teaching effectiveness. By employing data collection and deep mining techniques, the challenge of resource sparsity resulting from the original data is effectively mitigated, leading to enhanced recommendation algorithm accuracy. Through careful analysis of user feedback scores, similar data can be distinguished, enabling efficient information recommendation from English databases. To enhance the practical application of sharing intelligent teaching multimedia, the optimal gateway node serves as the gateway, thereby improving speed and stability. Future endeavors will focus on user segmentation and exploring the inclusion of learners' emotional classification to optimize the English teaching system.

### Funding

This work was supported by the Beijing University of Technology (Cultivation of Model Course Ideology and Politics Project in 2022, Grant No. KC2022SZ039) and the Beijing-Dublin International College at BJUT (Education and Teaching Research Project, Grant No. BDIC2022B02). The funders had no role in study design, data collection and analysis, decision to publish, or preparation of the manuscript.

### Grant Disclosures

The following grant information was disclosed by the authors:
Beijing University of Technology: KC2022SZ039.
Beijing-Dublin International College at BJUT: BDIC2022B02.

## Competing Interests

The authors declare there are no competing interests.

## Author Contributions

- Wangmeng Jiang conceived and designed the experiments, performed the experiments, analyzed the data, performed the computation work, prepared figures and/or tables, authored or reviewed drafts of the article, and approved the final draft.
- Qiang Ma conceived and designed the experiments, performed the experiments, analyzed the data, performed the computation work, prepared figures and/or tables, authored or reviewed drafts of the article, and approved the final draft.

## Data Availability

The code is available in the Supplemental File.

The data is available at Zenodo and the Connecticut Data Collaborative: http://data.ctdata.org/dataset/english-language-learners.

Sasha Cuerda. (2023). English Language Learners [Data set]. Zenodo. https://doi.org/10.5281/zenodo.7925050.

## Supplemental Information

Supplemental information for this article can be found online at http://dx.doi.org/10.7717/peerj-cs.1486#supplemental-information.

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
