# Peer review of "English teaching system design based on ID3 algorithm and node optimization"

_PeerJ Computer Science, doi:10.7717/peerj-cs.1486_

## Round 0.1 · original submission · Major Revisions

Couple of major improvements are needed as suggested by the experts, therefore we invite you to carefully update and resubmit.

Reviewer 1 ·

Basic reporting

An English multimedia teaching resource sharing platform is designed, which extracts feature items from multimedia teaching resources by ID3 information gain method and constructs a decision tree for resource push. In the process of resource sharing, structured peer-to-peer network is used to manage nodes, locate and share multimedia teaching resources in the system. In practical application, the platform can improve the sharing speed and utilization rate of teaching resources and realize accurate recommendation of English teaching resources. However, with the current quality, this article cannot be published. This article has many defects, so my suggestion is a minor revision.

Experimental design

No Comment

Validity of the findings

No Comment

Additional comments

(1) It is suggested that the author modify the title. The decision tree algorithm is too broad and should be replaced with the more representative "ID3 algorithm";
(2) In addition, the summary of the method in the title is not comprehensive, and a large amount of the paper introduces the recommendation of the collaborative filtering algorithm for multimedia English teaching resources for different users;
(3) One of the critical issues identified is the lack of explicit support and weak arguments to justify the proposed objectives, originality, and gaps that the study covers;
(4) I'm unsure of the contribution of this work overall - the motivation of the article should be constructed according to this comment. Please make it clear;
(5) As for the construction of English resource sharing platform, the author lacks a review and summary of relevant research methods, which leads to the practical significance of this work is not prominent enough;
(6) Use more professional English expressions, such as “When a node acts as an execution node, a trusted JVM, the Java Virtual machine, is supported, which sends information about application progress to reports.”
(7) In the process of resource recommendation, what is the classification of different learning resources, which will lead to a large difference in the score among users;
(8) In the actual test, the author uses P2P⁃BT platform (2015) as the reference, which makes the results non-extendable and adds some up-to-date methods for comparison.

Cite this review as

Reviewer 2 ·

Basic reporting

Strengthening the mining of educational information to improve the efficiency of data utilization of shared resources is the focus of the current research and thinking in the field of computer and education. The operating efficiency and network throughput of the English multimedia teaching resource sharing platform proposed in this paper are higher than those of the comparison platform. Maximize the use of teaching multimedia resources, avoid the waste of teaching multimedia resources.

The statement logic of the introduction is rather confusing. Is the author's research object "university education" or "vocational education"?
Is the "college English teaching resource library" created by the author or your team? I doubt this part contributes to the article.

The introduction should provide a more comprehensive explanation of these aspects to give the reader a better understanding of the study's significance.

Experimental design

Introduce the resource processing process in a more intuitive way (flow chart, number, etc.)

In the process of node optimization, what is the relationship between the comprehensive performance of resource nodes and node selection factors? This needs to be reflected in formula (3).

Validity of the findings

Delete some long-winded statements, “There is a problem in this method, such as setting the item that users evaluate not more than 1%, then the number of unrated items is more than 99%, and the similarity between them is more than 98%.”(Line 241-243).

The conclusion provides a good summary of the paper's contributions and future research directions.However, the authors could benefit from more explicitly highlighting the novelty and potential impact of their proposed method.

Cite this review as

---

## Round 0.2 · accepted · Accept

Consequent upon the recommendation of the experts in pleased to inform you that your paper is accepted, congratulations and thank you for your fine contribution.

Reviewer 1 ·

Basic reporting

The paper is improved with clear language, updated references and alligned with my previous comments. Now I have no further concerns and suggest the paper for publication.

Experimental design

Improved as per my previous comments.

Validity of the findings

Improved

Additional comments

No further comments and concerns.

Cite this review as

Reviewer 2 ·

Basic reporting

The authors have addressed all the raised concerns and the paper can be accepted in present form.

Experimental design

No Comment.

Validity of the findings

No Comment

Additional comments

No comment

Cite this review as